# Nano-Silicon Triggers Rapid Transcriptomic Reprogramming and Biochemical Defenses in *Brassica napus* Challenged with *Sclerotinia sclerotiorum*

**DOI:** 10.3390/jof9111108

**Published:** 2023-11-16

**Authors:** Qiuping Zhang, Jiaqi Wang, Jiajia Wang, Mulan Liu, Xiao Ma, Yang Bai, Qiang Chen, Song Sheng, Feng Wang

**Affiliations:** 1College of Agronomy, Hunan Agricultural University, Changsha 410128, China; zhangqiuping@hunau.edu.cn (Q.Z.); wjqhaha000728@163.com (J.W.);; 2Yuelushan Laboratory, Changsha 410128, China; 3Zhongshanshi Junyejiate Agricultural Technology Co., Ltd., Zhongshan 528400, Chinawangfenghif@126.com (Q.C.); 4College of Forest, Central South University of Forestry and Technology, Changsha 410004, China

**Keywords:** nano-silicon, *Brassica napus*, rapid transcriptomic reprogramming, biochemical defenses, *Sclerotinia sclerotiorum*

## Abstract

Stem rot caused by *Sclerotinia sclerotiorum* poses a significant threat to global agriculture, leading to substantial economic losses. To explore innovative integrated pest management strategies and elucidate the underlying mechanisms, this study examined the impact of nano-silicon on enhancing resistance to *Sclerotinia sclerotiorum* in *Brassica napus*. Bacteriostatic assays revealed that nano-silicon effectively inhibited the mycelial growth of *Sclerotinia sclerotiorum* in a dose-dependent manner. Field trials corroborated the utility of nano-silicon as a fertilizer, substantially bolstering resistance in the *Brassica napus* cultivar Xiangyou 420. Specifically, the disease index was reduced by 39–52% across three distinct geographical locations when compared to untreated controls. This heightened resistance was attributed to nano-silicon’s role in promoting the accumulation of essential elements such as silicon (Si), potassium (K), and calcium (Ca), while concurrently reducing sodium (Na) absorption. Furthermore, nano-silicon was found to elevate the levels of soluble sugars and lignin, while reducing cellulose content in both leaves and stems. It also enhanced the activity levels of antioxidant enzymes. Transcriptomic analysis revealed 22,546 differentially expressed genes in Si-treated *Brassica napus* post-*Sclerotinia* inoculation, with the most pronounced transcriptional changes observed one day post-inoculation. Weighted gene co-expression network analysis identified a module comprising 45 hub genes that are implicated in signaling, transcriptional regulation, metabolism, and defense mechanisms. In summary, nano-silicon confers resistance to *Brassica napus* against *Sclerotinia sclerotiorum* by modulating biochemical defenses, enhancing antioxidative activities, and rapidly reprogramming key resistance-associated genes. These findings contribute to our mechanistic understanding of Si-mediated resistance against necrotrophic fungi and offer valuable insights for the development of stem-rot-resistant *Brassica napus* cultivars.

## 1. Introduction

Stem rot, caused by the pathogen *Sclerotinia sclerotiorum*, is a pervasive agricultural disease that significantly impacts *Brassica napus*, commonly known as rapeseed [1]. Globally, rapeseed stands as the third-largest source of vegetable oil and a key protein meal, with its cultivation expanding in response to rising demand for food and renewable fuels [2,3,4,5,6,7].

In severe cases, the incidence rate of stem rot can reach up to 80%, leading to early plant withering, pod reduction, seed wrinkling, and a decrease in 1000-grain weight. Consequently, yield losses can range from 10% to 50% [8], translating to significant economic repercussions, with annual financial losses reaching into the billions of dollars [9].

Currently, the primary method of control relies on pesticides, which not only contribute to environmental pollution but also escalate production costs for farmers. Moreover, the unpredictable onset of *Sclerotinia sclerotiorum* makes timely pesticide application challenging, diminishing its efficacy. Over time, the pathogen has also developed resistance to these chemical treatments, further undermining their effectiveness.

*Sclerotinia sclerotiorum* is a broad-host-range necrotroph, capable of infecting over 600 plant species and causing rapid tissue maceration indiscriminately [10]. To date, no effective resistance sources have been identified, making it challenging to breed new varieties with high resistance to this pathogen. Recent research has employed omics studies, including genomics [11,12,13,14,15,16,17], transcriptomics [18,19], and proteomics [20,21], to identify disease resistance genes and understand the resistance mechanisms in rapeseed. Genetic engineering has also been utilized to introduce exogenous genes that can impair the infection ability of *Sclerotinia sclerotiorum* [22,23,24,25,26,27,28,29]. However, due to the complex pathogenicity of *Sclerotinia sclerotiorum* and the multifaceted resistance mechanisms in plants, a single gene approach has proven insufficient.

Given the limitations of current methods, there is an urgent need for environmentally friendly alternatives to manage diseases caused by *Sclerotinia sclerotiorum*. Silicon has been suggested as a viable option, as it has been shown to improve plant tolerance to various abiotic stresses [30], such as drought [31,32], salinity [33,34,35], heavy metals [36,37], high temperatures [38], freezing conditions [39], and ultraviolet radiation [40,41]. Moreover, silicon has demonstrated efficacy against biological stresses, including fungi, bacteria, and insect pests [42,43]. Previous studies have shown that silicon-treated plants exhibit enhanced resistance to diseases like powdery mildew in both wheat [44] and cucumber [45,46].

The mechanisms by which silicon enhances plant defense have been extensively studied. Silicon contributes to the formation of physical barriers that impede pathogen penetration, including the wax layer, cuticle, and cell wall [47,48,49]. Specifically, silicon aids in the silicification of leaf epidermis cells, the formation of cutin–silicon double-layer structures, and the induction of papillae that prevent pathogenic infections [50,51,52]. Additionally, silicon activates biochemical defense mechanisms, increasing the activities of protective enzymes and inducing the synthesis of secondary metabolites that bolster plant resistance [53,54,55,56,57,58,59,60,61].

While extensive research has been conducted on the role of silicon in improving plant resistance to diseases like rice blast, bacterial blight, and powdery mildew in cucumber and melon, limited studies have focused on its potential in preventing stem rot in *Brassica napus.* This study aims to evaluate the effects of silicon on mitigating stem rot caused by *Sclerotinia sclerotiorum* in rapeseed through both field and laboratory tests, while also investigating the underlying physiological and molecular mechanisms.

## 2. Materials and Methods

### 2.1. Plant Material and Silicon Application

A winter rapeseed variety, Xiangyou 420, was planted in fields at three locations in China, including Changsha (112°58′42″ E, 28°11′49″ N), Hengyang (112°24′10″ E, 27°0′58″ N), and Anxiang ( 112°10′2″ E, 29°24′47″ N) by mechanical planters in the 2019/2020 growth season for a field disease survey. For the laboratory test, the rapeseed plants inoculated with *Sclerotinia sclerotiorum* were planted in 25 cm pots containing soils mixed with fertilizer in a controlled growth chamber under 450 μmol m^−2^ s^−1^ light intensity with a 16 h photoperiod provided by fluorescent and incandescent lamps. The temperature was maintained at 25 °C in the light and 20 °C in the dark, and the humidity was 60%. A silicon solution (Si content ≥ 60 g/L, pH value 6–8, Na content ≤ 5 g/L, water insoluble content ≤ 10 g/L, produced by Guangdong Zhongshan Junye Jiate Co., Ltd., Zhongshan, China, amount ≤ 10 g/L) was sprayed at the seedling stage, early florescence, and the blossom period, respectively. The control treatments were sprayed with water.

### 2.2. Inoculation with Sclerotinia sclerotiorum

In this study, *Sclerotinia sclerotiorum* was isolated from field-sown canola plants [62]. A sterilized toothpick was inserted into potato dextrose agar medium and co-cultured with mycelium of *Sclerotinia sclerotiorum* in culture bottles in an incubator at 22 °C until covered with mycelium.

The toothpick co-cultured with *Sclerotinia sclerotiorum* was inoculated on the main stem of rapeseed after 5 d of silicon treatment during the blossom period, and water treatment instead of silicon was chosen as the control for comparative analysis. The sampling was a randomized design with three biological replicates, with five time points (0, 1, 2, 3, and 5 days post-inoculation [dpi]). The humidity of the chamber was 90–95% during the inoculation. Three leaves from the inoculated plants were detached after 0, 1, 2, and 5 dpi. The tissues harvested from one biological replicate at each time point were pooled as one sample and were frozen immediately in liquid nitrogen and stored at −80 °C for an enzyme assay and measurement of the content of oxygen free radicals (OFR), H_2_O_2_, and malondialdehyde (MDA).

### 2.3. Measurement of Disease Index

During the harvest period, 20 sites were selected from the three fields, and 10 plants were analyzed in each site, giving a total of 600 plants for analysis.

The classification of the stem disease degree was recorded (grade 0: no disease; grade 1: less than 1/3 branches are diseased or lesion on main stem does not exceed 3 cm; grade 2: 1/3–2/3 branches are diseased or lesion on main stem exceeds 3 cm; grade 3: more than 2/3 branches are diseased or lesion on middle and lower part of main stem exceeds 3 cm). The disease index was calculated according to the following formula:$$I=\frac{\sum \left(d_{i}\times l_{i}\right)}{L\times 3}\times 100$$

Here, in the formula, *I* is the disease index, *d_i_* is the value of each severity grade and *l_i_* is the number of diseased plants of each severity grade, while *L* means the total number of diseased plants investigated.

### 2.4. Antagonistic Experiments

Potato dextrose agar (PDA) medium is commonly utilized for the cultivation of *Sclerotinia sclerotiorum*. In order to investigate the impact of nano-silicon on the growth of this pathogen, solutions of nano-silicon at varying concentrations were integrated into the PDA medium. Specifically, the PDA was prepared with nano-silicon concentrations of 50, 75, and 100 mg/mL. These amended media were then dispensed into sterile 10 cm petri dishes under aseptic conditions in a laminar flow cabinet. A control was established using PDA medium devoid of nano-silicon to serve as a benchmark for comparative analysis. Subsequently, a mycelial plug of *Sclerotinia sclerotiorum*, measuring 0.5 cm in diameter, was aseptically transferred to the center of each petri dish containing the nano-silicon-amended and control PDA media. Each treatment, including the control, was replicated three times. The inoculated dishes were then incubated at a constant temperature of 20 °C for a period of five days. Post-incubation, the growth of the mycelium was quantified by measuring the diameter of the fungal colony.

### 2.5. Enzyme Assay

Canola leaves treated with silicon and inoculated with *Sclerotinia sclerotiorum* were sampled at 0, 1, 2, 3, and 5 days post-inoculation (dpi) to assay the activities of superoxide dismutase (SOD), peroxidase (POD), ascorbic acid peroxidase (APX), polyphenol oxidase (PPO), and catalase (CAT). Leaves treated with water served as controls. For each time point and treatment, three biological replicates were processed. Leaf samples (0.1 g) were homogenized in 1 mL of 50 mM phosphate buffer (pH 7.0) and centrifuged, and the supernatant was used for enzyme activity assays as per established methods [63,64,65,66,67].

### 2.6. Measurement of H_2_O_2_ and MDA Content

The content of hydrogen peroxide (H_2_O_2_) and malondialdehyde (MDA) in canola leaves was monitored at the same time points post-inoculation with the pathogen. Leaves sprayed with clear water acted as controls. Three replicates per treatment were analyzed using a hydrogen peroxide content kit and spectrophotometric absorption at 532 nm for H_2_O_2_ [67], and the method of Castrejón [68] was used for MDA content determination.

### 2.7. Measurement of Oxygen Free Radical Content

The content of oxygen free radicals (OFR) in the leaves was quantified using the Elstner method [69]. Fresh leaf tissue (0.1 g) was homogenized in a phosphate buffer and subjected to ultrasonic disruption. The homogenate was centrifuged, and the supernatant was reacted with xanthine oxidase and xanthine, followed by sulfanilic acid and α-naphthylamine solutions. After chloroform extraction, the absorbance at 530 nm was measured. Three replicates were used for each treatment, and the OFR content was calculated against a standard curve.

### 2.8. Measurement of Mineral Elements and Soluble Sugar, Lignin, and Cellulose

The mineral element and soluble sugar, lignin, and cellulose content of canola treated with silicon and water from Changsha, Hengyang, and Anxiang were compared. The contents of K, Na, and Ca in the stems and leaves of rapeseed variety Xiangyou 420 were determined by ICP-AES [70], while Si content was measured by alkali fusion [71]. Additionally, the content of soluble sugar, lignin, and cellulose was determined by the colorimetric method of anthrone and sulfuric acid, sulfuric acid acidolysis titration, and anthronecolorimetry, respectively [72,73].

### 2.9. RNA-Seq Analysis

For the RNA-Seq analysis, raw reads in the fastq format were initially assessed for quality using the FastQC program (https://www.bioinformatics.babraham.ac.uk/projects/fastqc/, accessed on 13 July 2023)) to evaluate parameters such as Q20, Q30, GC content, and sequence duplication levels. Subsequently, the data were processed for read alignment using the Hisat2 software (version 2.2.1) [74]. De novo alignment was conducted using the Trinity tool (https://github.com/trinityrnaseq/trinityrnaseq/wiki, accessed on 15 July 2023) provided by BMK Corporation (details undisclosed). The aligned reads were then normalized by converting them to fragments per kilobase of transcript per million fragments mapped (FPKM) to determine gene and transcript expression levels. Custom R scripts were employed to analyze gene expression patterns, and heatmaps were generated using a combination of the ggplot2, reshape2, gplots, and dplyr packages in R (version 4.1.2).

### 2.10. Construction of Gene Co-Expression Networks via WGCNA

To construct gene co-expression networks, weighted gene co-expression network analysis was used [75,76]. The analysis parameters were set as follows: power Estimate 28, maxBlock Size 25,000, min Module Size 30, and merge Cut Height 0.25. A module phenotype correlation heatmap was generated using the ggplot2 package in R. Subsequently, network analysis and visualization were performed using BioSciTools (https://bio.tools/bioscitools, accessed on 8 August 2023). Within the “green” module, the “Network Plot” function was executed under default settings. Hub genes within this module were selected based on GS1 and datKME values exceeding 0.95.

## 3. Results

### 3.1. Enhancement of Stem-Rot Resistance in Brassica napus by Silicon Application

To investigate the impact of nano-silicon on the resistance of *Brassica napus* to stem rot, we conducted field trials in three different locations: Anxiang, Changsha, and Hengyang. The disease index for stem rot in the rapeseed variety Xiangyou 420 was assessed. The average disease indices for silicon-treated and control plants were 5.0 and 10.6 in Anxiang, 6.7 and 11.1 in Changsha, and 2.9 and 5.4 in Hengyang, respectively (Figure 1). These findings suggest that silicon application significantly enhances the stem-rot resistance of Xiangyou 420. In laboratory tests, lesions appeared in the control plants at 1 day post-inoculation (dpi), whereas they were observed in silicon-treated plants at 2 dpi (Figure 2a). The diameters of the disease spots in silicon-treated plants measured 2.7 mm and 7.3 mm at 2 dpi and 5 dpi, respectively, compared to 4.6 mm and 13.3 mm in the control plants (Figure 2b). These laboratory results corroborate the field data, indicating a consistent silicon-mediated enhancement in stem-rot resistance. Antagonistic assays revealed marked differences in the growth of *Sclerotinia sclerotiorum* in response to varying concentrations of nano-silicon after 5 days of culture (Figure 3). In the absence of nano-silicon, the mycelium of *Sclerotinia sclerotiorum* was essentially overgrown in the Petri dish. However, when 100 mg/mL of nano-silicon was added to the medium, mycelial growth was completely inhibited. Additionally, a dose-dependent reduction in the average diameter of mycelial growth was observed as the external silicon concentration increased (Figure 3). In summary, these results demonstrate that the application of silicon has a positive effect on enhancing resistance to *Sclerotinia sclerotiorum* in rapeseed.

### 3.2. Effects of Silicon on Potassium (K), Calcium (Ca), and Sodium (Na) Content in Stems and Leaves of Rapeseed

To elucidate the impact of silicon application on the absorption of essential nutrients associated with pathogen resistance in the Xiangyou 420 cultivar, we conducted a comparative analysis of nutrient element content in the leaves and stems between silicon-treated and control plants. Our findings revealed that the silicon content in both leaves and stems of the treated plants was significantly higher compared to the controls across all three field trials (Figure 4a). In Changsha, the calcium (Ca) content in the leaves and stems of Xiangyou 420 was also significantly elevated compared to the control plants; however, this trend was not observed in Anxiang and Hengyang (Figure 4b). Excluding Changsha, the potassium (K) content in the leaves and stems of silicon-treated plants in Anxiang and Hengyang was notably higher than that in the control plants (Figure 4c). Conversely, sodium (Na) content in the leaves and stems of silicon-treated plants was significantly reduced compared to the controls (Figure 4d). These results suggest that elevated potassium levels in the leaves and stems of silicon-treated plants may contribute to enhanced resistance against stem rot.

### 3.3. Effects of Silicon on Soluble Sugar, Lignin, and Cellulose Levels in Leaves and Stems of Rapeseed

The concentrations of soluble sugar, lignin, and cellulose in the leaves and stems of the Xiangyou 420 cultivar from Anxiang, Changsha, and Hengyang were quantitatively assessed. A comparative analysis was conducted between the silicon-treated plants and the untreated controls (Figure 5). For soluble sugar, its levels in the leaves and stems of silicon-treated plants were significantly higher than those in the control plants across all three field trials, with the exception of leaves from the Changsha location (Figure 5a). Similarly, lignin concentrations in the leaves and stems of silicon-treated plants were notably elevated compared to the controls in all three locations (Figure 5b). Conversely, cellulose levels exhibited an inverse trend. The concentrations in the leaves and stems of silicon-treated plants were significantly lower than those in the control plants across the three field trials, although the difference was not statistically significant for stems in the Anxiang location (Figure 5c). Based on these findings, it is posited that the increased levels of soluble sugar and lignin in the leaves and stems of silicon-treated plants may contribute to enhanced resistance against stem rot.

### 3.4. Effects of Silicon on Antioxidant Enzyme Activities in Rapeseed following Inoculation with Sclerotinia sclerotiorum

To elucidate the physiological and biochemical mechanisms underlying silicon-induced resistance to stem rot in the Xiangyou 420 cultivar of *Brassica napus*, we assessed the temporal variations in the activities of five key antioxidant enzymes—superoxide dismutase (SOD), peroxidase (POD), polyphenol oxidase (PPO), catalase (CAT), and ascorbate peroxidase (APX)—in both silicon-treated and control plants. These measurements were conducted at multiple time points: 0, 1, 2, 3, and 5 days post-inoculation (dpi). Our findings revealed that the activities of SOD (Figure 6a), POD (Figure 6b), CAT (Figure 6d), and APX (Figure 6e) were significantly elevated at 1 dpi, reaching their peak levels at 2 dpi before subsequently declining at 3 dpi. Conversely, PPO activity exhibited an increase at 2 dpi, peaking at 3 dpi, and then diminishing at 5 dpi (Figure 6c). Importantly, silicon treatment led to a marked augmentation in the activities of all five enzymes under investigation. By 5 dpi, the activities of POD, PPO, CAT, and APX in the leaves of the silicon-treated plants surpassed those observed in the control plants (Figure 6). In summary, the application of silicon significantly enhanced the activities of antioxidant enzymes, thereby contributing to the improved resistance of *Brassica napus* to stem rot.

### 3.5. Effects of Silicon on OFR, H_2_O_2_, and MDA Content in Rapeseed following Inoculation with Sclerotinia sclerotiorum

To assess the extent of oxidative damage and its temporal variations in the Xiangyou 420 cultivar post-inoculation with *Sclerotinia sclerotiorum*, we quantified the levels of oxygen free radicals (OFR), hydrogen peroxide (H_2_O_2_), and malondialdehyde (MDA) in both silicon-treated and control plants at various time points following inoculation. Our findings revealed that the OFR content in both the silicon-treated and control plants exhibited a similar trend: it increased gradually, peaking at 2 days post-inoculation (dpi), before subsequently declining (Figure 7a). However, the OFR levels in the silicon-treated plants were consistently lower than those in the control plants post-inoculation (Figure 7a). In contrast, the temporal changes in H_2_O_2_ content diverged from those of OFR (Figure 7b). Specifically, the H_2_O_2_ content in the silicon-treated plants peaked at 2 dpi and then decreased, whereas in the control plants, it reached its maximum value at 1 dpi (Figure 7b). Notably, the H_2_O_2_ levels in the silicon-treated plants were higher than those in the control plants after 1 dpi (Figure 7b). As for MDA content, we observed a progressive increase over time in both treatment groups. However, the MDA levels in the silicon-treated plants were consistently lower than those in the control plants at all measured time points (Figure 7c). In summary, these results suggest that the silicon-treated Xiangyou 420 plants experienced less oxidative damage compared to the control plants, as evidenced by lower levels of OFR and MDA and altered H_2_O_2_ dynamics.

### 3.6. Transcriptomic Analysis Unveils Rapid and Robust Silicon-Induced Differential Gene Expression in Brassica napus Challenged with Sclerotinia sclerotiorum

To explore the gene expression landscape in *Brassica napus* in response to *Sclerotinia sclerotiorum* infection, we performed an in-depth RNA-Seq analysis on both control (CK) and silicon-treated plants at 1 and 5 days post-inoculation (dpi) (The transcriptome data has been submitted to the SRA database with the accession number of PRJNA1025336, Appendix A). The initial principal component analysis (PCA) yielded distinct clusters among the four sample groups. The first principal component (PC1) accounted for 42.99% of the total variance, while the second principal component (PC2) explained an additional 27.23%. Subsequent analysis of differential gene expression revealed a total of 22,546 differentially expressed genes (DEGs) across all experimental conditions. Intriguingly, the most substantial subset of DEGs emerged at 1 dpi, consisting of 6290 DEGs when comparing control and silicon-treated samples (Figure 8). This observation underscores a rapid and robust plant response to silicon treatment. Given the extensive catalog of DEGs, we employed weighted gene co-expression network analysis (WGCNA) to further refine our investigation. This approach allowed us to identify functional gene modules that are specifically associated with silicon-induced resistance.

### 3.7. Identification and Functional Classification of Hub Genes Correlated with Silicon-Mediated Disease Resistance

In a comprehensive gene co-expression network analysis using WGCNA (Figure 9), a total of 22,546 differentially expressed genes (DEGs) were classified into 14 distinct modules with varying numbers of genes. Notably, the green module, comprised of 3328 genes, displayed the highest correlation (r = 0.91, *p*-value = 5 × 10^−5^) to the phenotypic trait of spot size, signifying its potential role in disease resistance mechanisms. Subsequent network plotting using the default settings yielded 45 hub genes within the green module that possessed both GS1 and datKME values exceeding 0.95. Further functional annotation sorted these hub genes into five main functional groups: Group 1, focusing on transcriptional and post-translational regulation, includes genes such as “Zinc finger, RING/FYVE/PHD-type” and “Zinc finger, C_2_H_2_-like”, which are postulated to modulate gene expression responses to silicon treatment for disease resistance; Group 2, associated with signal transduction and membrane proteins, involves genes like “Tubby C-terminal-like domain” and “Tetraspanin, conserved site”, potentially affecting silicon perception and activation of resistance pathways; Group 3, comprising enzymes and metabolic genes, contains “SGNH hydrolase-type esterase domain” and “Protein phosphatase 2C”, among others, which are speculated to modify cellular metabolism mechanisms pivotal for silicon-induced disease resistance; Group 4 encompasses genes related to defense and stress response, including “Hs1pro-1, N-terminal” and “EF-Hand 1, calcium-binding site”, which could directly participate in cellular defense mechanisms; and lastly, Group 5, with genes of unknown functions such as “Protein of unknown function DUF581” and “Domain of unknown function DUF250”, may uncover novel silicon-triggered resistance pathways. In summary, these identified hub genes represent key components involved in silicon-mediated signal perception and downstream regulatory mechanisms, thus warranting further in-depth investigation (Figure 10).

## 4. Discussion

Stem rot, caused by the necrotrophic fungus *Sclerotinia sclerotiorum*, poses a significant threat to oilseed rape (*Brassica napus*), resulting in substantial yield losses globally. Building upon the existing body of research that underscores the role of silicon in enhancing plant resilience to various stressors, our study offers compelling evidence that nano-silicon application significantly fortifies *Brassica napus* against *Sclerotinia sclerotiorum* infection under both field and controlled conditions. In this study, we demonstrated that nano-silicon application enhanced resistance against the necrotrophic pathogen *Sclerotinia sclerotiorum* in *Brassica napus* under both natural field conditions and controlled inoculation in the phytotron. *Brassica napus* treated with nano-silicon exhibited significantly reduced disease severity compared to control plants without silicon application. We posit that silicon uptake and deposition in plant tissues likely restricted the spread and colonization of *Sclerotinia sclerotiorum* mycelia, prolonging the latency period required for full pathogen establishment and thus delaying disease onset.

*Sclerotinia sclerotiorum* infection triggers a hypersensitive response in host plants, resulting in a metabolic oxidative burst and production of cytotoxic reactive oxygen species (ROS) that eventually cause programmed cell death. As a defense strategy, host plants activate a repertoire of endogenous antioxidant enzymes to neutralize and scavenge excess oxygen radicals to prevent uncontrolled tissue damage. Key enzymes involved in this antioxidant response include superoxide dismutase (SOD), peroxidase (POD), polyphenol oxidase (PPO), catalase (CAT), and ascorbate peroxidase (APX). Their activities directly reflect the plant’s capacity to mitigate oxidative stress under pathogen attack. Previous studies showed differential activities of peroxidase, polyphenol oxidase, phenylalanine ammonia lyase, and superoxide dismutase in resistant versus susceptible *Brassica napus* genotypes, before and after *Sclerotinia* inoculation [77]. More resistant *Brassica napus* cultivars possessed greater enzymatic capabilities to detoxify oxygen radicals, preventing runaway cell death and consequently delaying disease expansion [21]. In our current study, exogenous silicon application significantly augmented the antioxidant system in *Brassica napus* by modulating the activities of POD, SOD, PPO, CAT, and APX post-inoculation with *Sclerotinia*. This was evidenced by the reduced accumulation of OFR and MDA in nano-silicon-treated plants compared to non-treated controls. Taken together, these results indicate that silicon likely enhances the activities of endogenous antioxidant enzymes to effectively scavenge excess ROS produced during *Sclerotinia* infection, thereby improving overall disease resistance.

In addition to modulating plant antioxidative capacity, silicon appears to bolster resistance by altering the absorption and distribution of essential mineral nutrients linked to defense responses, such as calcium, potassium, and sodium. Previous research has shown that optimal calcium and potassium levels can increase plant tolerance to various abiotic stresses, as well as enhance resistance against pathogen infection [78,79]. However, excessive accumulation of sodium can negatively impact plants by interfering with potassium uptake and distribution and antagonizing vital calcium signaling pathways [80]. Our results revealed that exogenous silicon fertilization significantly increased the absorption and accumulation of beneficial macronutrients calcium and potassium in the leaves and stems of *Brassica napus*, while concurrently reducing sodium content across all field trials.

In Changsha, leaf and stem calcium levels were markedly higher in nano-silicon-treated plants versus non-treated controls. Although not significant in Anxiang and Hengyang, calcium has been previously shown to bolster plant defenses by regulating oxidative bursts, defense gene expression, and downstream protein phosphorylation cascades during pathogen infection [81]. On the other hand, potassium content was notably elevated in the leaves and stems of silicon-treated *Brassica napus* in both Anxiang and Hengyang. Moderate potassium concentrations can mitigate disease incidence in plants through mechanisms such as promoting carbon metabolism and increasing sugar reserves [82,83,84], which are closely linked to defense responses [85]. Accordingly, we observed a concomitant increase in potassium and total soluble sugar levels in the leaves and stems of nano-silicon-treated *Brassica napus* across trials.

Cell wall fortification is another major mechanism by which silicon enhances physical barriers and mechanical strength against pathogen invasion. Lignin and cellulose are two integral cell wall components influencing disease resistance. Enhanced lignin deposition and the maintenance of lignin polymer integrity have proven crucial for impeding fungal penetration and restricting *Sclerotinia* colonization [86]. Lignin reinforces secondary cell walls against physical breaches while impeding the ingress and diffusion of degradative enzymes and phytotoxins secreted by invading pathogens [87]. Lignin biosynthesis also yields phenolic compounds that can additionally inhibit fungal reproduction and spread [88]. Our results demonstrated that nano-silicon fertilization markedly increased stem and leaf lignin content while decreasing cellulose levels in *Brassica napus* across all three field locations. This cell wall fortification likely forms effective physical barriers, restricting Sclerotinia access and thus bolstering resistance.

To further probe the molecular mechanisms activated by silicon in *Brassica napus* during Sclerotinia infection, we performed global transcriptomic profiling at two key time points post-inoculation. Intriguingly, the extensive RNA-Seq dataset revealed that nano-silicon triggered rapid and dramatic transcriptional reprogramming as early as one day post-inoculation, consisting of over 6000 differentially expressed genes between silicon-treated and control plants. Such substantial and swift molecular changes underscore the vital function of silicon as an early warning signal to preemptively potentiate plant defense responses prior to disease onset.

Downstream co-expression network analysis enabled us to pinpoint 45 hub genes distributed among five functional groups that appear to orchestrate the silicon-activated signaling cascades and cellular reprogramming underlying enhanced disease resistance. Zinc finger proteins likely act as central transcriptional regulators, akin to *OsWRKY45* in rice [89], to modulate the expression of critical silicon-responsive genes. Perception and transduction of the exogenous silicon signal is postulated to involve membrane proteins like Tetraspanin, known to assemble signaling complexes in model plants like Arabidopsis [90]. These, in turn, could relay the signal to metabolic genes such as phosphatases, which can reconfigure the cellular metabolism essential for defense responses [91]. Other genes are directly involved in cellular protection, including calcium-binding EF-Hand proteins that transduce extracellular silicon cues to initiate downstream defense mechanisms, as observed in Arabidopsis [92]. Additionally, the unknown functional roles of other hub genes present exciting opportunities to uncover previously undiscovered facets of silicon-mediated resistance. Collectively, these identified candidates appear to play a highly coordinated and complex role in sensing silicon availability and manifesting the appropriate genetic reprogramming to bolster plant defenses. While their protective actions were elucidated here in the context of *Brassica napus*, similar mechanisms may be functionally conserved across diverse silicon-responsive crop species susceptible to *Sclerotinia sclerotiorum* diseases.

## 5. Conclusions

This study provides novel insights into silicon’s multifaceted protective effects against the devastating necrotrophic pathogen *Sclerotinia sclerotiorum* in *Brassica napus*. Our integrated field, biochemical, and molecular analyses shed light on the intricate signaling pathways, antioxidative mechanisms, and cell-wall fortifications activated by silicon to augment plant resistance. The rapid and robust transcriptional changes induced by silicon highlight its vital function as an early signal for potentiating defense responses prior to infection. The panel of 45 identified hub genes offers promising targets for better elucidating and exploiting silicon’s prophylactic potential to control *Sclerotinia sclerotiorum* diseases. Given the lack of complete genetic resistance, integrating proper silicon nutrition into integrated pest management programs could provide durable, broad-spectrum protection against this pathogen in oilseed rape and other vulnerable cash crops. Further efforts to validate the functional roles of these genes through mutational studies and translate key findings to optimal field application rates and timing are warranted. Overall, this systematic study significantly advances our mechanistic understanding of the multiple modes of silicon-induced resistance in the *Brassica napus–Sclerotinia sclerotiorum* pathosystem, providing a foundation to improve silicon-based control measures against stem rot.

## Figures and Tables

**Figure 1 jof-09-01108-f001:**
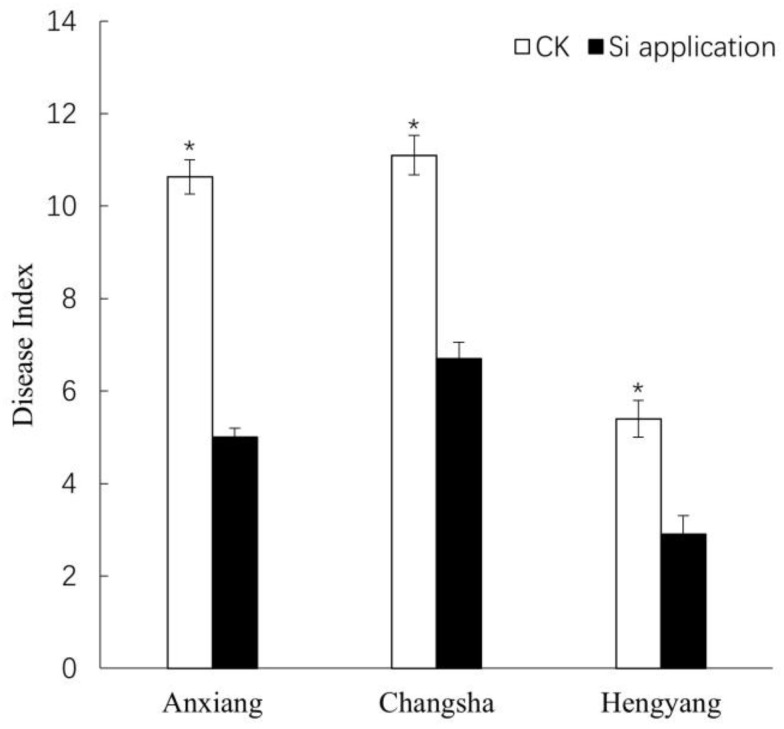
Disease indices of stem rot in Xiangyou 420 in the field trials in Anxiang, Changsha and Hengyang. Data are means ± SD, biological replicates (*n* = 3). * indicates that the differences between silicon-treated plants and the controls (CK) are significant (*p* < 0.05).

**Figure 2 jof-09-01108-f002:**
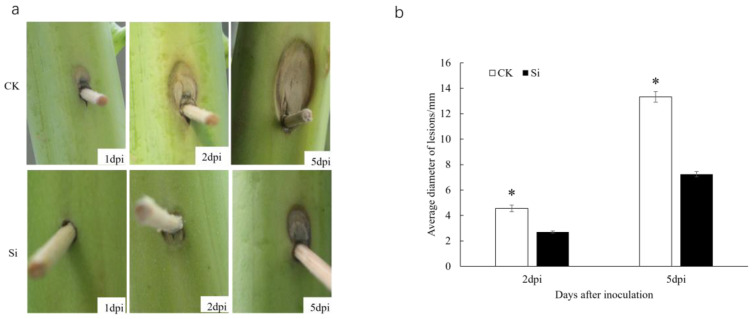
Spot sizes after inoculation of *Sclerotinia sclerotiorum* in main stems of Xiangyou 420. (**a**) Disease spots on rape stems after inoculation with *Sclerotinia sclerotiorum*; (**b**) Statistical analysis results of lesion diameters. Note: dpi means days after inoculation, Data are means ± SD, biological replicates (*n* = 3). * indicates that the differences between silicon-treated plants and the controls (CK) are significant (*p* < 0.05).

**Figure 3 jof-09-01108-f003:**
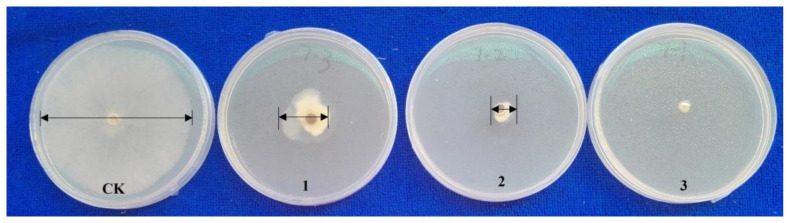
Mycelial growth of *Sclerotinia sclerotiorum* in culture media with different silicon concentrations. 1–3 indicate that the culture medium contained 100, 75, and 50 mg/mL nano-silicon, respectively. CK was without nano-silicon application.

**Figure 4 jof-09-01108-f004:**
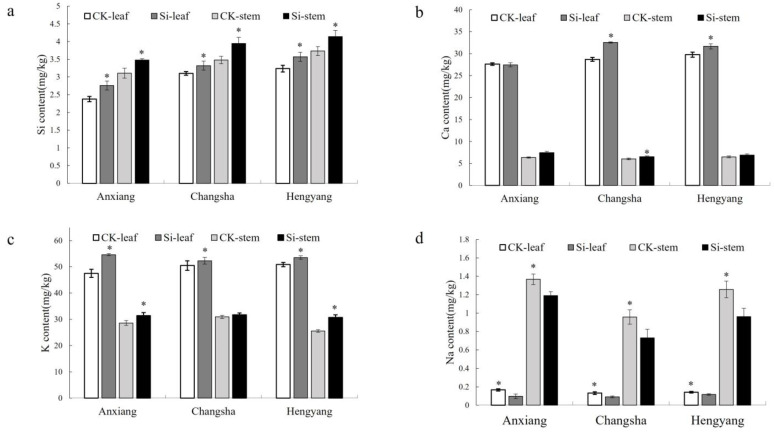
The content of K, Na, Ca, and Si in the stems and leaves of the Si-treated and control plants of Xiangyou 420. Figures (**a**–**d**) indicate the Si, Ca, K, and Na content in the stems and leaves of canola, respectively. Data are means ± SD, biological replicates (*n* = 3). * indicates that the differences between the silicon treatment and the CK are significant at the level of 0.05.

**Figure 5 jof-09-01108-f005:**
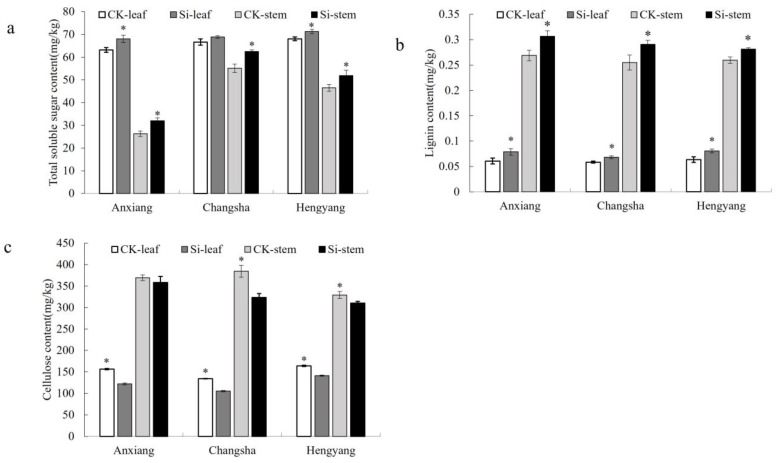
The content of soluble sugar, lignin, and cellulose in the stems and leaves of Si-treated and control plants of Xiangyou 420. Figures (**a**–**c**) indicate the content of total soluble sugar, lignin, and cellulose, respectively. * indicates significant differences between the silicon treatment and the CK at the level of 0.05.

**Figure 6 jof-09-01108-f006:**
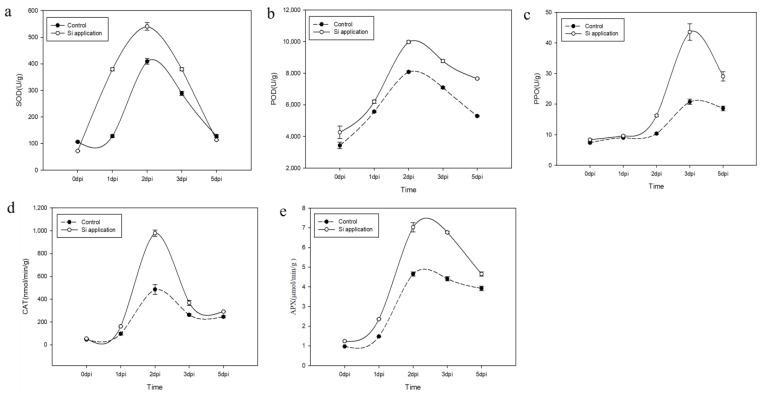
Activities of oxidation-related enzymes in the Si-treated and control plants of Xiangyou 420 after *Sclerotinia sclerotiorum* inoculation. Figures (**a**–**e**) indicate the content of SOD, POD, PPO, CAT, and APX in canola stems and leaves, respectively. Data are means ± SD, biological replicates (*n* = 3).

**Figure 7 jof-09-01108-f007:**
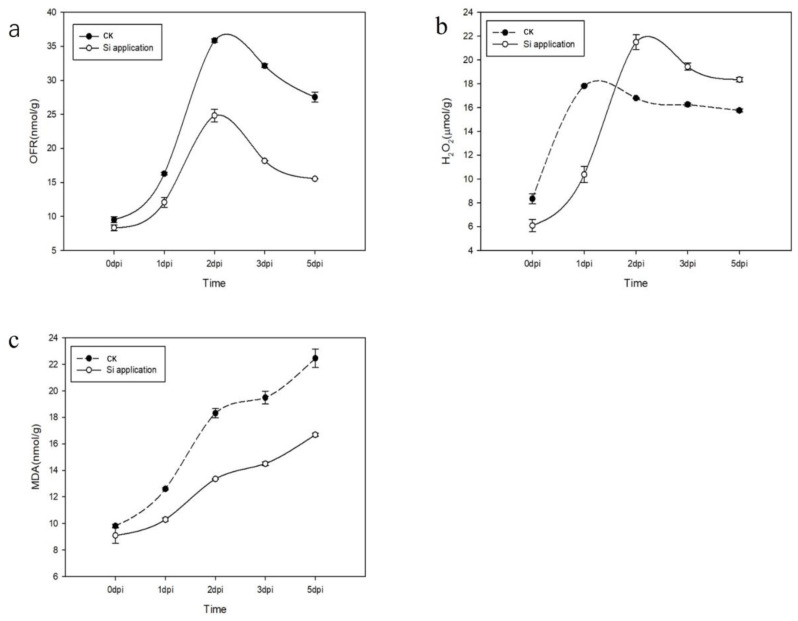
The content of OFR (**a**), H_2_O_2_ (**b**), and MDA (**c**) in leaves of Si-treated and control plants of Xiangyou 420 after *Sclerotinia sclerotiorum* inoculation. (**a**) OFR content; (**b**) H_2_O_2_ content; (**c**) MDA content. Data are means ± SD, biological replicates (*n* = 3).

**Figure 8 jof-09-01108-f008:**
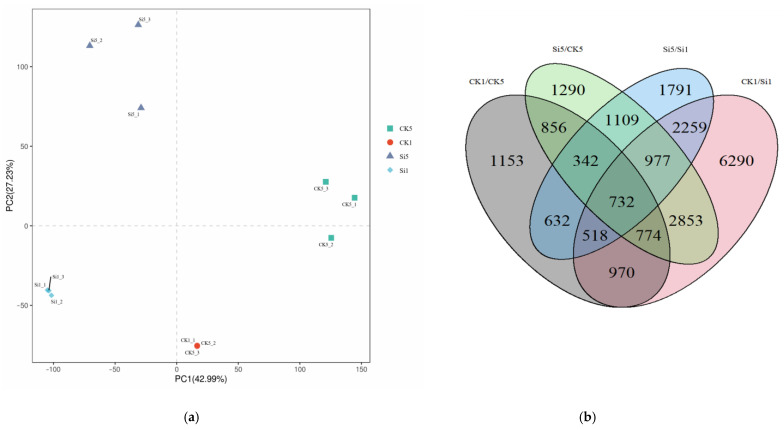
The principal component analysis (PCA) and differentially expressed genes (DEGs) in leaves of the Si-treated and control plants. (**a**) scatter plot of principal component analysis (PCA) for Si-treated and control plants RNA-seq datasets, the axes (PC1 and PC2) represent the principal components that explain the maximum amount of variance in the data, the different symbols (square, circle, triangle, and diamond) and colors represent different categories or classifications within the dataset; (**b**) Venn diagram of DEGs, the numbers in each section of the Venn diagram represent the number of observations that fall into the corresponding categories represented by each circle, the overlaps between the circles represent the observations that are shared between the categories.

**Figure 9 jof-09-01108-f009:**
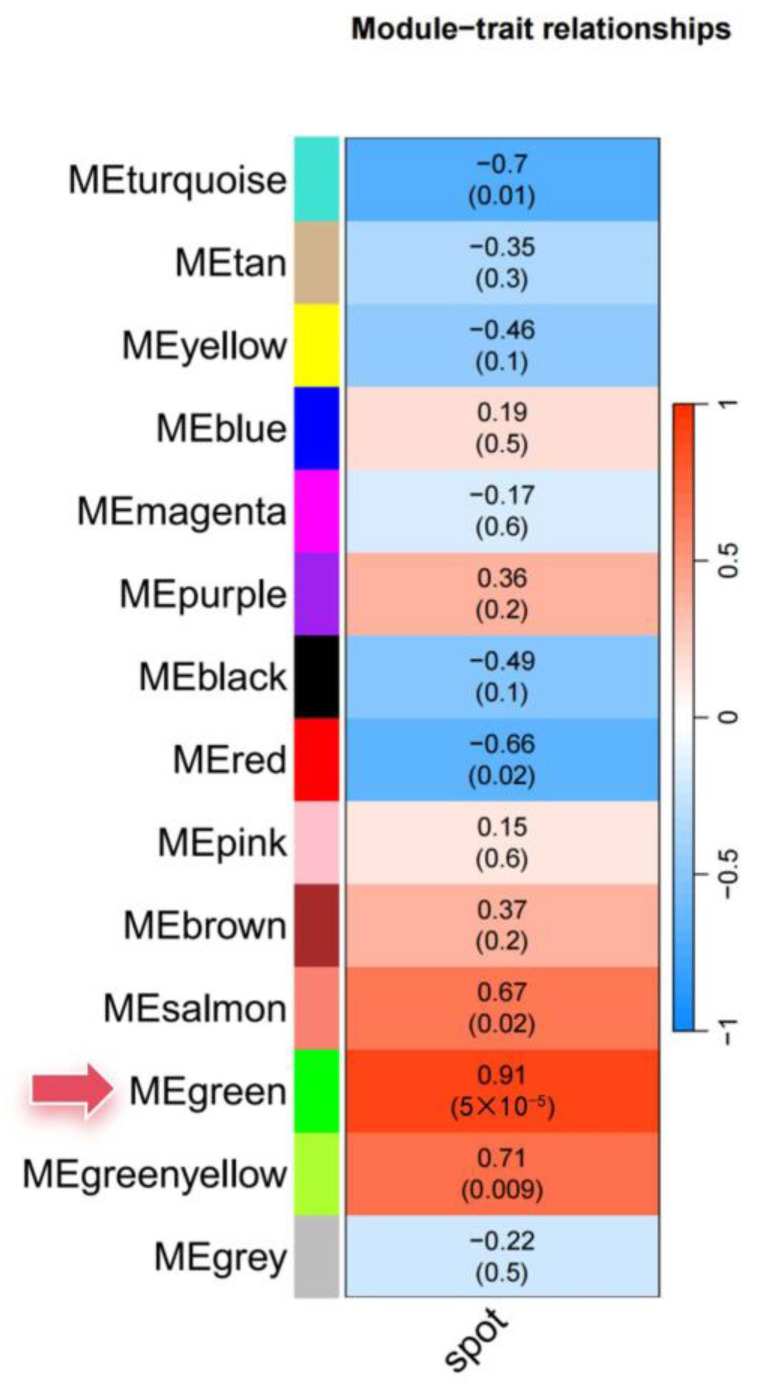
Comprehensive gene co-expression network analysis using WGCNA.

**Figure 10 jof-09-01108-f010:**
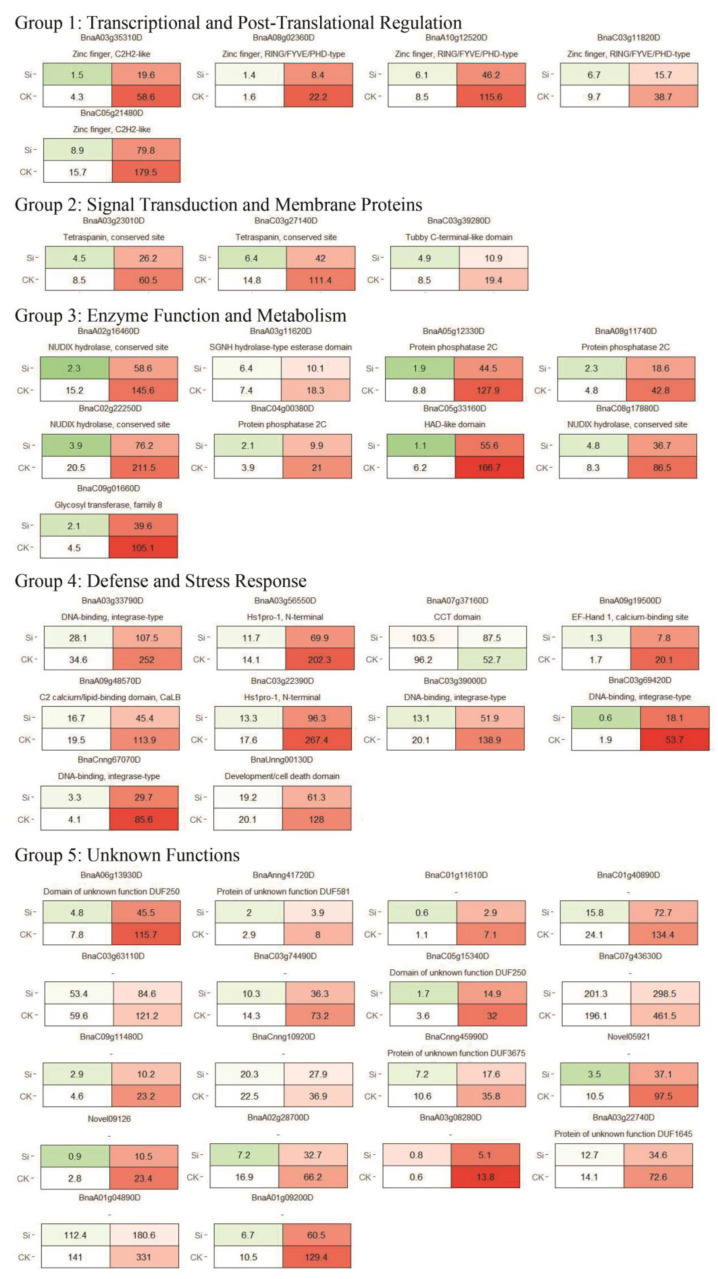
Functional annotation and classification of hub genes.

## Data Availability

Data are contained within the article.

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
