# Peer review of "Nano-Silicon Triggers Rapid Transcriptomic Reprogramming and Biochemical Defenses in Brassica napus Challenged with Sclerotinia sclerotiorum"

_jof, 2023, doi:10.3390/jof9111108_

Round 1

Reviewer 1 Report

Comments and Suggestions for Authors

Dear Editor

The article shows how silicon can be utilized to inhibit pathogenic fungi Sclerotinia sclerotiorum in challenged rapeseed plants, as well as the underlying anti-pathogenic defense mechanism. To validate the findings, the authors used not just microbiological approaches but also transcriptome analysis. The article is improved version than the previous but still need  major improvement.

The formatting is not according to the journal.

It would be better to use a software to arrange bibliography/references.

Line 16: fungi name italic please; similarly line 45; 56;60; 193; 214

Line 18: scientific name italic please; similarly line 25; 29; 40; 77-79; 262

Line 83: space between Xiangyou 420; line 124; 126

Please check if the disease index formula is in the format of the journal.

Section 2.4 is unclear on how the author performed the antifungal analysis; it will be better to give a reference along with the clearly mentioned methodology for the antifungal analysis.

Section 2.5: it is unclear#

 on what samples were enzymatic assays performed? Were they treated with fungus? Were they performed on field samples or green house samples?

Mention full form of MDA first then use short form in the manuscript.

What were the controls for section 2.5, 2.6 and 2.7?

How may replication were used in experiments mentioned in section 2.4, 2.5, 2.6 and 2.7?

The font is different in line 150-151? Check the font requirements of the journal.

Reference for section 2.8?

Improve the quality of figure 2

Improve the quality of figure 3

improve the quality of figure 4

improve the quality of figure 5

Comments on the Quality of English Language

improved

Reviewer 2 Report

Comments and Suggestions for Authors

The manuscript is a study of the physiological and defensive responses of B. napus against S.s.

However I have recommendations that need to be corrected.

In the introduction, the authors must color the commercial and global value of the cultivation of B. nappus, as well as its trends over time (years) as its commercial importance increases worldwide.

Similarly, they must include in the introduction the commercial value (in dollars) of the losses caused by S.S.

Throughout the entire manuscript, the scientific name of Sclerotium is incorrect, it is a serious mistake on the part of the authors.

Methods

Why did they use so few experimental replicates in the field? Because the n=3 reported in the physiological figures are very few (Figure 1).

In Fig. 1, put the names of the sites studied

The white (CK) must be in white bar and not in black, and those that used Silicon in dark or gray colors.

Fig. 2 b, the axes are not noticeable, they are too small

Fig.3 the culture plates must go in the CK direction (left), and the concentrations on the right are poorly organized.

Fig. 4 and 5, the colors of the bars are poorly organized, CK should be in white and the rest in black and gray colors, not in bars with complex squares.

Fig. 6, 7 the axes cannot be read and it is difficult to analyze the information.

Fig 8 A, PCA is not understood, due to the smallness of the points

Fig 9. I don't understand that figure, what are the color legends, what genes or interactions they are showing, I didn't understand.

A table that analyzes the interactions of the genes and what they are would be better, I refer to Fig 10.

Comments on the Quality of English Language

None

Round 2

Reviewer 1 Report

Comments and Suggestions for Authors

The revised manuscript by the authors is a much-improved version compared to the previous version but still needs further improvement before publication in the Journal of Fungi.

- The formatting is not according to the journal format. Cross-check the font of the material and methods section and results section. (Line 87 -110 and Line 189 onwards).

- It is still unclear about the controls used in the sections 2.5, 2.6 and 2.7.

- there is no mention about the replications used in the sections 2.4-2.7

Reviewer 2 Report

Comments and Suggestions for Authors

NA

Comments on the Quality of English Language

NA

Author Response

Thank you very much for the review of our revised manuscript.